

# Tunable dispersion of the edge states in the integer quantum Hall effect

**Maik Malki**[1*]**, Götz S. Uhrig**[1◇]

**1** Lehrstuhl für Theoretische Physik 1, TU Dortmund University, Germany

⋆ maik.malki@tu-dortmund.de    ◇ goetz.uhrig@tu-dortmund.de

## Abstract

Topological aspects represent currently a boosting area in condensed matter physics. Yet there are very few suggestions for technical applications of topological phenomena. Still, the most important is the calibration of resistance standards by means of the integer quantum Hall effect. We propose modifications of samples displaying the integer quantum Hall effect which render the tunability of the Fermi velocity possible by external control parameters such as gate voltages. In this way, so far unexplored possibilities arise to realize devices such as tunable delay lines and interferometers.


# 1 Introduction

## 1.1 General context

Subjecting a two-dimensional electron gas at low temperature to a strong perpendicular magnetic field results in the well-known quantization of the transverse conductivity $\sigma_{xy} = \nu e^2/h$ with $\nu \in \mathbb{N}$ which is called integer quantum Hall effect [1] (IQHE). The remarkable high-precision with which the integer quantum Hall conductivity can be measured is attributed to its relation to topological invariants [2–7]. Shortly after the discovery of the IQHE another topological effect was measured and baptized the fractional quantum Hall effect [8, 9] since Hall plateaus appear at fraction filling factors $\nu$. The discovery of the integer and fractional quantum Hall effect triggered a steadily growing interest in topological phenomena in condensed matter phenomena.

The IQHE is a single-particle phenomenon [10, 11]; no interaction between the electrons needs to be taken into account which facilitates its understanding greatly. In the bulk, the interpretation of the IQHE is that the filling factor $\nu$ equals the total Chern number of the filled Landau bands. This Chern number is a topological invariant [3–5] related to the fundamental Berry phase [12]. This warrants the high precision of resistance measurements fulfilling Ohm's law without any non-linear corrections [13].

A closer understanding is gained if one realizes that the actual charge currents are carried by gapless edge states [6] which cross the Fermi level. They have to exist at the boundaries because the Chern number jumps across them [14]. The number of gapless edge states [2] corresponds to the Chern number $\nu$. Each of these edge states can be seen as single-channel conductor [15] propagating only in one direction along the edge which therefore are called chiral edge states. They allow for adiabatic transport [16] because backscattering is forbidden which makes such transport particularly interesting for applications.

It is fascinating that the IQHE can be put into a larger context of Chern insulators [14] which need not be induced by external magnetic fields. Complex kinetic Hamilton operators on lattices, i.e., complex hoppings, can imply non-trivial Chern numbers and concomitant edge modes as they appear in the IQHE. The seminal example is the Haldane model [17]. Its quantum Hall effect is called anomalous because no magnetic field is required. The inclusion of the spin degree of freedom [18, 19] opens the possibility of the quantum spin Hall effect [20–22]. The quantum anomalous Hall effect [23–25] as well as the quantum spin Hall effect [26] have been realized experimentally.

## 1.2 Present objective

For clarity, we focus here on the IQHE and do not take the spin into account which is left to future research. The topological protection of the chiral edge states and the complete suppression of backscattering in these edge states suggests that the chiral edge states enable robust applications. Calibrating resistance standards to extremely high precision is certainly a wonderful example. Yet, in the present study we want to trigger research on *further* applications.

We will investigate the Fermi velocity $v_\mathrm{F}$ occurring in the chiral edge states. It represents the group velocity of electrical signal transmitted through the system. Hence, it determines the speed of signal transmission. If it can be tuned it can be used to influence the time signals need to cross the sample. In this way, certain delays can be imposed and used for signal processing, for instance for interference measurements. We emphasize that the Fermi velocity does not influence the widely studied DC conductivity which is not the quantity of interest in our study, in contrast to the majority of theoretical studies in the literature.

Triggered by the observation that the Fermi velocity of edge states in Chern insulators on lattices differs depending on the details of the edges [27] a systematic study of modifications

of the edges of the generic Chern insulator in the Haldane model revealed that the Fermi velocity can indeed be tuned over orders of magnitude by changing external parameters such as gate voltages [28]. The key idea is to modify the edges by decorations such that local levels are created which are brought in weak contact with the dispersive edge modes. The ensuing hybridization leads to a weakly dispersing mode of which the Fermi velocity can be tuned by changing the energy of the local modes. If the local levels are in resonance with the edge modes the sketched mechanism is at work and a low Fermi velocity appears. If they are out-of-resonance the hybridization is ineffective and the edge states remain strongly dispersive. The tuning of the local decorated edge modes can be achieved by gate voltages.

This fundamental idea has been carried over from the spinless Haldane model to the spinful Kane-Mele model [29]. In this study, the effect of disorder in the decorated Haldane model has been addressed as well. It was shown that the Fermi velocity is robust against weak disorder if the dispersion is not too flat, i.e., if the Fermi velocity is not too low. Hence, in contrast to the naive expectation of complete robustness due to the topological origin of the edge states disorder changes the dispersion of the modes and can deteriorate signal transmission *beyond* the DC conductivity.

As pointed out in the general context, tunable Fermi velocities open the possibility of interesting applications such as delay lines or interference devices. Unfortunately, the lattice systems known so far cannot yet be tailored on the nanoscale to render the experimental verification of the theoretical proposal possible. So far, solid state systems postulated by density-functional theory can be envisaged to yield realizations in the future [30–32]. Alternatively, intricate optical lattice may make proof-of-principle realizations of tunable Fermi velocities possible [33, 34]. Yet, the search for different realizations is called for. In particular, the high standard of designing nanostructures in semiconductor systems suggests to look for such systems for the realization of tunable dispersions of edge states.

This brings us back to the IQHE which is based on a semiconducting interface generating a two-dimensional (2D) electron gas and a perpendicular magnetic field. If one is able to tailor the boundaries of the 2D electron gas in a way that mimics the decoration of 2D lattice models tunable Fermi velocities become possible. Indeed, it has been proposed by one of us that attaching bays to the boundaries of a Hall sample allows us to generate local modes in the bay [28]. If they are slightly opened to the 2D bulk a weak hybridization is realized and the physics established so far for lattice systems should carry over to the IQHE. The basic geometry is sketched in Fig. 1.

Currently, it is possible to implement bays in the submicrometer range in IQHE samples. For instance, a single-electron source has been realized by coupling a quantum dot to a 2DEG via quantum point contacts and a gate voltage setting the dot potential [35]. An additional gate voltage at the quantum point contacts is used to control the transmission, see Fig. 1(b) so that the hybridization can be tuned as indicated in Fig. 1(c). If such a coupled quantum dot is repeated periodically the geometry in Fig. 1(a) is obtained. This proposed setup will be studied in the sequel as an exemplary model for the realization of tunable Fermi velocities in the IQHE.

Below, we present calculations showing that the Fermi velocity $v_F$ can be tuned by adding periodically arranged bays to an integer quantum Hall sample. The paper is organized as follows. In Sect. 2 we specify the model Hamiltonian describing the IQHE and the numerical approach to compute the edge states and their dispersion. Sect. 3 illustrates step by step how the spectrum of the decorated IQHE is structured. In particular, we focus on the effects of the hybridization between the modes in the bays and the edge modes because this is the mechanism altering the Fermi velocities. The results for tuned Fermi velocities are presented in Sect. 4. Finally, Sect. 5 collects our findings and provides an outlook.

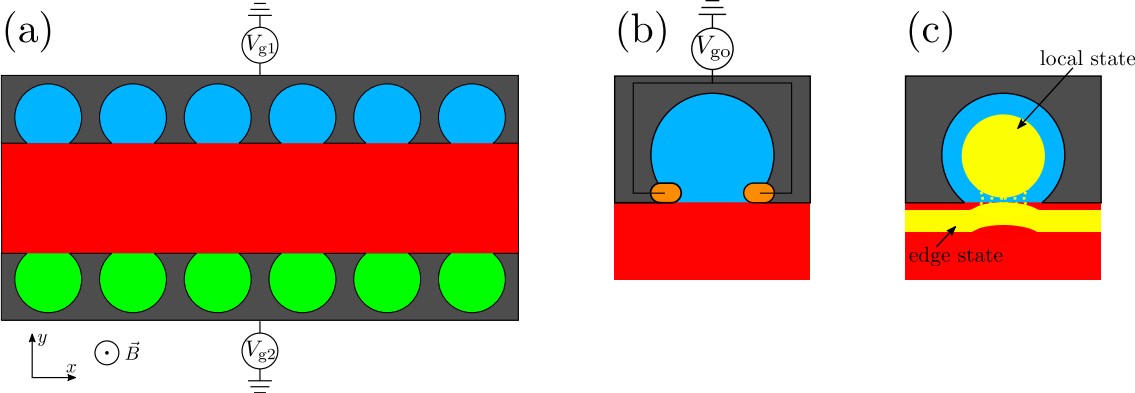

Figure 1: Panel (a): proposal of a decorated quantum Hall sample with tunable Fermi velocity. A perpendicular magnetic field puts the two-dimensional electron gas in the quantum Hall phase. Two independent gate voltages $V_{g1}$ and $V_{g2}$ change the potential of the blue bays at the upper boundary and of the green bays at the lower boundary, respectively. The grey area is inaccessible to the electrons. The size of the opening of the bays to the bulk 2DEG can be controlled by a gate voltage $V_{go}$ as depicted in panel (b). The size of the opening controls the degree of hybridization of the local mode within the bays and the edge mode in the 2D bulk, see panel (c).

## 2  Model and technical aspects

The present work is designated to illustrate the tunability of the Fermi velocity on a proof-of-principle level. For the sake of clarity, we assume that the upper and the lower boundary are sufficiently far away from each other so that the edge states localized at the upper and at the lower boundary do not influence each other. Practically, this means that the magnetic length $\ell_B = \sqrt{\hbar/(|eB|)}$ is significantly smaller than the width $L_y$ of the quantum Hall sample, i.e., the external magnetic field must be large enough. Then, it is not necessary to study a system of which both boundaries are decorated. Hence, we focus here on a sample with quadratic bays at the upper boundaries, but no decoration at the lower boundary which is kept smooth. The precise shape of the bays does not matter for our proof-of-principle calculations. Within the colored area shown in the panels of Fig. 2 the electrons can move freely. Their dynamics is only governed by their kinetic energy. The boundaries are supposed to be infinitely hard walls as indicated by thick black lines.

Applying a perpendicular magnetic field in $z$-direction, see panel (a) in Fig. 2, is incorporated in the usual way by minimal coupling

$$\vec{p} \to \vec{p} - q\vec{A}, \tag{1}$$

where the charge reads $q = -|e|$ and $\vec{A}$ is the magnetic vector potential. No electron-electron interactions are considered so that the full Hamilton operator reads

$$H = \frac{1}{2m}\left(\vec{p} - q\vec{A}\right)^2, \tag{2}$$

where $m$ is the (effective) mass of the electrons. The electrons are confined to the $xy$-plane; we do not consider their spin degree of freedom. This can be justified because the two spin species $\uparrow$ and $\downarrow$ are decoupled in the perpendicular magnetic field [36, 37].

Due to the translational invariance in the $x$-direction a Landau gauge is particularly appropriate. We choose the Landau gauge in $x$-direction $\vec{A} = B(-y, 0, 0)$ so that the momentum $k_x$

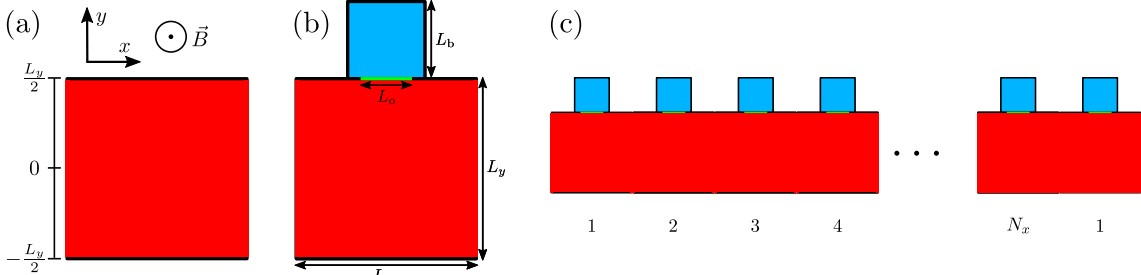

Figure 2: Sketch of the considered geometries of increasing complexity. Panel (a) displays the standard IQHE sample without any decoration of the boundaries; its width is denoted by $L_y$ and its total length by $L_x$. Periodic boundary conditions in $x$-direction are assumed. Panel (b) shows the single unit considered where the dimensions of the bay and the coupled bulk are given. Note the opening of the bay shown in green; its width is denoted by $L_o$. The total sample consists of $N_x$ such units as shown in panel (c) so that $L_x = N_x L_{xp}$.

remains manifestly conserved. This leads to the continuum Hamilton operator

$$H_{\text{bulk}} = \frac{\hbar^2}{2m}\left[\left(-\mathrm{i}\frac{\partial}{\partial x} + \frac{qB}{\hbar}y\right)^2 - \frac{\partial^2}{\partial y^2}\right] \tag{3a}$$

$$= \frac{m\omega_c^2}{2}\left(y + \mathrm{i}\ell_B^2\frac{\partial}{\partial x}\right)^2 - \frac{\hbar^2}{2m}\frac{\partial^2}{\partial y^2} \tag{3b}$$

in real space where we use the definition of the cyclotron frequency $\omega_c = |e|B/m$ and the magnetic length $\ell_B = \sqrt{\hbar/|eB|}$. It is implied that $x$ and $y$ take only values in the colored regions of the panels in Fig. 2 unless stated otherwise.

## 2.1 Bulk system

Solving the Hamiltonian (3b) in case of a bulk system without any boundaries leads to the famous Landau levels with quantized energy values [38]

$$E_n = \hbar\omega_c(n + 1/2), \ n \in \mathbb{N}. \tag{4}$$

The corresponding wave functions are plane waves in $x$-direction and Gaussians multiplied with Hermite polynomials in $y$-direction

$$\psi(n, k_x, y) = N\mathrm{e}^{-(y-y_0)^2/2\ell_B^2}H_n((y-y_0)/\ell_B)\mathrm{e}^{\mathrm{i}k_x x} \tag{5}$$

because the Hamiltonian corresponds to shifted harmonic oscillators in $y$-direction. The wave functions are normalized by $N$, $H_n$ is the $n$th Hermite polynomial, and $y_0 = k_x\ell_B^2$ determines the center of the wave function $\psi(n, k_x, y)$ in $y$-direction. These facts about the bulk Landau level will be helpful for the understanding of the more complicated situations and serve as reference. Below, we consider more and more details of the actual model depicted in Fig. 2(c).

## 2.2 Sample of finite width $L_y$

Next, we consider a sample as shown in Fig. 2(a), i.e., of finite width in $y$-direction, but with translational invariance along $x$ due to periodic boundary conditions. A numerical treatment is required which we introduce here. It is chosen flexible enough to be extended subsequently to the decorated sample including the bays.

For simplicity we set the effective electron mass $m = 1$, Planck's constant $\hbar = 1$, and use $B$ henceforth for $|e|B$. This amounts to setting $\omega_c = 1$, i.e., to using $\hbar\omega_c$ as energy unit. The resulting bulk Hamiltonian reads

$$H_{\text{bulk}} = \frac{1}{2}\left[\left(\frac{y}{\ell_B^2} + i\frac{\partial}{\partial x}\right)^2 - \frac{\partial^2}{\partial y^2}\right]. \tag{6}$$

As displayed in Fig. 2(a) the boundary conditions in the $y$-direction imply $V(y) = \infty$ for $|y| \geq L_y/2$. We use the same Landau gauge as before in the bulk system. In $x$-direction, we exploit the translational invariance using the plane wave ansatz

$$\psi(x, y) = \exp(ik_x x)\psi(y). \tag{7}$$

This leads to the Hamilton operator which acts on $\psi(y)$

$$H_{\text{undec. con.}} = \frac{1}{2}\left[\left(\frac{y}{\ell_B^2} - k_x\right)^2 - \frac{\partial^2}{\partial y^2}\right], \tag{8}$$

with $|y| \leq \frac{L_y}{2}$. We tackle this problem by discretizing the $y$ coordinate by a mesh with distance $a$ between the points. It is understood that $a$ is much smaller than any other physical length scale in the system, i.e., $\ell_B$ and $L_y$. The resulting model resembles a tight-binding model, but we emphasize that its discrete character is just due to the approximate treatment of the continuum. We make sure that the discretization mesh is always fine enough so that the results are close to the continuum values, see below.

So the discretized Hamiltonian, expressed in second quantization, which approximates the continuum operator (8) reads

$$H_{\text{undec. dis.}} = \sum_y \left[\frac{1}{2}\left(\left(\frac{y}{\ell_B^2} - k_x\right)^2 + \frac{5}{2a^2}\right)c_{y,k_x}^\dagger c_{y,k_x} - \frac{2}{3a^2}c_{y+a,k_x}^\dagger c_{y,k_x}\right.$$
$$\left. + \frac{1}{24a^2}c_{y+2a,k_x}^\dagger c_{y,k_x} + \text{h.c.}\right] - \frac{1}{24a^2}c_{b(y),k_x}^\dagger c_{b(y),k_x}, \tag{9}$$

where $c_{y,k_x}$ ($c_{y,k_x}^\dagger$) annihilates (creates) an electron with wave vector $k_x$ in $x$-direction at coordinate $y$. To this end, the second derivative is approximated by the difference quotient

$$\frac{\partial^2\psi(y)}{\partial y^2} \approx \frac{1}{a^2}\left[-\frac{1}{12}\psi(y-2a) + \frac{4}{3}\psi(y-a) - \frac{5}{2}\psi(y) + \frac{4}{3}\psi(y+a) - \frac{1}{12}\psi(y+2a)\right]. \tag{10}$$

This formula cannot be applied to values of $y$ which are close to a boundary because the values $\psi(y+a)$ and $\psi(y+2a)$ may not exist, see Fig. 3. In fact, if $y_{\text{bdry}}$ is the value right at the boundary (red site partly in the shaded area in Fig. 3), then $\psi(y_{\text{bdry}}) = 0$ holds due to the hard-wall boundary condition and one does not need a terms at $y_{\text{bdry}}$. What is needed is an approximation of the second derivative at $y_{\text{bdry}} - a$ for which $\psi(y_{\text{bdry}} + a)$ is required. One could simply omit this term, but this omission would introduce an error of the order of $a$ with respect to the continuum situation which we intend to approximate. Hence, we exploit that $\psi(y_{\text{bdry}}) = 0$ and that a continuous function can be approximated by its Taylor expansion around $y_{\text{bdry}}$. In linear order this implies $\psi(y_{\text{bdry}} + a) \approx -\psi(y_{\text{bdry}} - a)$ which leads the last term in (9) where we used the symbol $b(y) = y_{\text{bdry}} - a$ for the value of $y$ adjacent to the boundary. This improves the results roughly by one order in $a$, especially at the important edges of the sample.

In this way, we can very accurately compute eigen energies of the plain Hall sample as function of $k_x$. In particular, we obtain the wanted dispersion of the edge states. The level of

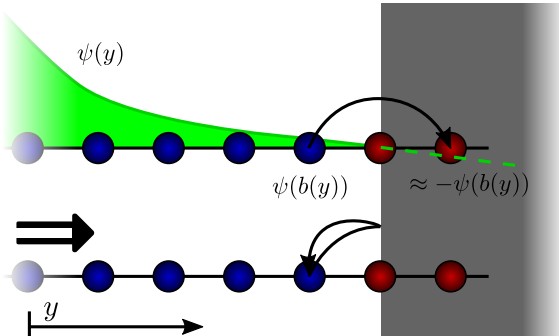

Figure 3: Illustration of the approximation used in immediate vicinity of a boundary in order to improve the approximation of the continuous system by a discretized one, see main text.

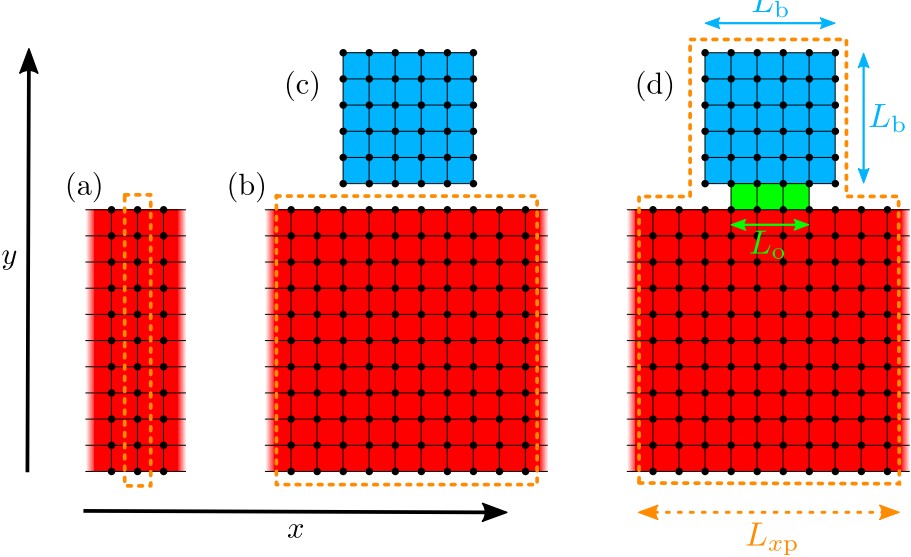

Figure 4: Sketches of the meshes used to capture the physics of the quantum Hall sample with and without bays. The strip geometry (a) and the rectangular geometry (b) are used to describe the sample without bays. The decoupled bay (c) is considered to compute the energy spectrum of the isolated bay as reference for the coupled system shown in panel (d). The orange dashed lines indicate the respective unit cells.

complexity is illustrated by Fig. 4 where the discretization meshes are shown. The calculation for the plain sample without any decoration only requires to discretize the $y$-axis, shown in Fig. 4(a), because the other spatial dependence is fully captured by the plane wave ansatz (7). This can be done very efficiently because only a relatively small number of sites is required. But in order to be able to later include the bays as shown in Fig. 4(d) we first re-calculate the sample without bays by considering the grid in panel (b).

## 2.3 Fully discretized samples

Enlarging the unit cell as shown in panel Fig. 4(b) leads to the continuum Hamilton operator

$$H_{(b)} = \frac{1}{2}\left(\frac{y^2}{\ell_B^4} + 2i\frac{y}{\ell_B^2}\frac{\partial}{\partial x} - \frac{\partial^2}{\partial x^2} - \frac{\partial^2}{\partial y^2}\right), \tag{11}$$

with the periodic condition for the wave function

$$\psi(x + L_{\text{xp}}, y) = \exp(ik_x L_{\text{xp}})\psi(x, y). \tag{12}$$

We stress that this condition allows us to determine the value of $k_x$ only up to multiplies of $2\pi/L_{\text{xp}}$, as usual if a reduce unit cell in real space is considered.

In the approximate discretized system the first order derivatives are expressed by using the central finite difference quotient in forth order accuracy

$$\frac{\partial \psi(x, y)}{\partial x} \approx \frac{1}{a}\left[ \frac{1}{12}\psi(x - a2, y) - \frac{2}{3}\psi(x - a, y) + \frac{2}{3}\psi(x + a, y) - \frac{1}{12}\psi(x + 2a, y) \right] \tag{13}$$

wherever possible. Close to a hard-wall boundary the value $\psi(x + 2a, y)$ is not known because it refers to sites outside of the considered domain. Then this term is simply omitted. The improvement used for the second derivative based on the mirroring explained in Fig. 3 cannot be used at hard walls in $x$-direction because the resulting correction terms would be local densities with imaginary prefactors spoiling the hermitecity of the Hamiltonian.

Thus, the Hamiltonian $H_{(\text{b})}$ is discretized in both directions. Expressed in second quantization it is given by

$$
\begin{aligned}
H = \sum_{x,y} &\left[ \frac{1}{2}\left( \frac{y^2}{\ell_B^4} + \frac{5}{a^2} \right)c_{x,y}^\dagger c_{x,y} - \frac{2}{3a^2}c_{x,y+a}^\dagger c_{x,y} + \frac{1}{24a^2}c_{x,y+2a}^\dagger c_{x,y} + \frac{\text{i}2By}{3a}c_{x+a,y}^\dagger c_{x,y} \right. \\
&\left. - \frac{\text{i}By}{12a}c_{x+2a,y}^\dagger c_{x,y} + \text{h.c.} \right] - \frac{1}{24a^2}c_{x,b(y)}^\dagger c_{x,b(y)} - \frac{1}{24a^2}c_{b(x),y}^\dagger c_{b(x),y},
\end{aligned}
\tag{14}
$$

where $x$ and $y$ run over the discrete sites within the colored areas in Fig. 4. The very last term occurs at hard-wall boundaries in $x$-direction, i.e., treating the bays, improving the second derivatives. The periodicity condition (12) carries over to

$$c_{x+L_{\text{xp}},y} = c_{x,y}\text{e}^{\text{i}k_x L_{\text{xp}}} \tag{15}$$

in second quantization.

The Hamiltonian (14) can be used to numerically calculate the spectrum for any shape of the integer quantum Hall sample. We employ it below to consider the finite strip without bays first , cf. Fig. 4(b), and isolated bays, cf. Fig. 4(c), for reference purposes. Finally, we pass on to the coupled system, cf. Fig. 4(d). Then, we also have to include the effect of the gate voltages, see Fig. 1. Gate voltage $V_{\text{go}}$ controls the size of the opening. This is implemented in our calculation by the choice of the geometry, i.e., by the value of $L_{\text{o}}$. Since we only consider bays at the upper boundary there is no $V_{g2}$ to study. The gate voltage $V_{g1}$ is implemented by the Hamiltonian part

$$H_{\text{bays}} = -\sum_{x,y \in \text{bays}} V_{g1}c_{x,y}^\dagger c_{x,y} \tag{16}$$

where we incorporated the value of the charge into $V_{g1}$, i.e., we use $V_{g1}$ for $|e|V_{g1}$.

For small values of $a$ the Hamiltonian (14) corresponds to very large, though sparsely populated matrices. We do not need all eigen values of them because we focus on the energies of the lowest Landau level up to about the third Landau level. In particular, the high-lying eigenvalues are strongly influenced by the discretization and hence they are meaningless for the underlying continuum model. In order to handle the diagonalization within given intervals of the spectrum efficiently for large sparse matrices we employ the FEAST eigen value solver. The FEAST algorithm [39] uses the quantum mechanical density matrix representation and counter integration techniques to solve the eigenvalue problem within a given search interval. Now, we are in the position to calculate the dispersion of the lowest eigen states and thus also able to calculate the Fermi velocities being the derivatives of the dispersion at the Fermi level.

# 3 Dispersions in decorated quantum Hall samples

So far, we analyzed the Landau levels in the bulk, see Sect. 2.1, and we introduced the approximate Hamiltonians to describe hard-wall boundaries of varying shapes, see Sects. 2.2 and 2.3. Here we present the results for geometries of increasing complexity. First, we address the strip geometry, i.e., the sample without any bays. Then, we study the isolated bays before we address the full coupled system, cf. Fig. 4. For clarity, we focus on the lowest Landau levels.

## 3.1 Strip geometry

In the case of a hard-wall confining potential in $y$-direction, i.e., $V(y) = \infty$ for $|y| > L_y/2$, one still expects to find eigen values and eigen states bearing similarities to the bulk solutions. For instance, the eigen function exponentially localized in the middle of the strip hardly feel the hard-wall confining potential. Hence they closely resemble the bulk functions (5) and their energies are exponentially close to the bulk Landau levels (4), see also below.

Moreover, the lowest eigen functions localized right at the boundary, i.e., $k_x = \pm L_2/2\ell_B^2$, equal the eigen function of the second Landau level $n = 1$. This is so because the zero of the antisymmetric wave functions coincides with the boundary [40] as is well known from the text book problem in quantum mechanics of a parabolic potential cut off at its apex by an infinite potential step. Thus, the antisymmetric Hermite polynomials are solutions which satisfy the boundary condition where they are localized. The influence of the other boundary is exponentially small if $\ell_B \ll L_y$ which is the limit we presuppose. These special points are used to verify the accuracy of the calculations based on the discretized model Hamiltonian in comparison to the continuum solutions.

For the discretized description to approximate the continuum efficiently in $y$-direction, the distance $a$ between sites must be small enough to capture the dependence of the Hermite polynomials (5) on $y$. Since $H_n(y)$ has $n$ zeros on the root mean square length $\ell_B\sqrt{n+1/2}$ we arrive at the constraint

$$a \ll \ell_B \frac{\sqrt{n+1/2}}{n+1} \approx \frac{\ell_B}{\sqrt{n+1}}. \tag{17}$$

In $x$-direction the wave length set by $2\pi/k_x$ sets an upper limit of $a$ so that we have to require

$$a \ll \frac{2\pi}{k_x}. \tag{18}$$

While (17) needs to be fulfilled in all our calculations, (18) is not required in the solution of (9), i.e., if the system in Fig. 4(a) is considered, but only if the fully discretized model introduced in Sect. 2.3 is considered.

In addition to these numerical requirements, we argued that we want to consider the case where the edge states at the upper and at the lower boundaries do not interfere. This requires

$$\ell_B \frac{\sqrt{n+1/2}}{n+1} \ll L_y \tag{19}$$

on physical grounds. The left hand side is the root mean square of the spatial extension of the $n$th Landau level in $y$-direction. We focus on the lowest bands anyway so that $n = 0$ and $n = 1$ are the relevant cases. For concreteness, we henceforth use the values $\ell_B = 1\mu$m, $a = 0.01\ell_B$ and $L_y = 10\ell_B$. These values are in accordance with the above considerations for numerical accuracy and independence (up to exponentially small corrections) of the edge states.

Considering the mesh in $y$-direction depicted in Fig. 4(a) we obtain the results (blue solid curves) shown in Fig. 5 where they are compared to the bulk results (4) (red dashed lines). Clearly, for small wave vectors one obtains flat bands agreeing very well with the bulk Landau

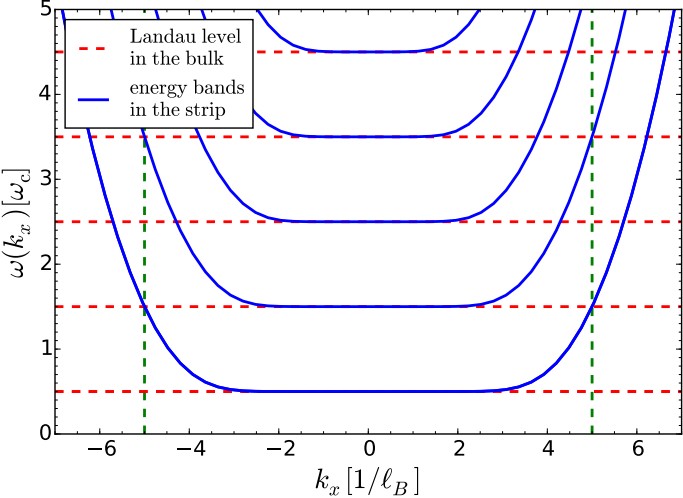

Figure 5: The blue curves show the dispersions of the Landau levels of a strip of finite width $L_y$, see Fig. 1(a). The red dashed lines indicate the equidistant energy spectrum of the Landau levels in the bulk. The vertical dashed lines are located at $k_x = \pm L_y/2\ell_B^2$ indicating the states which are localized at the upper and lower boundaries of the sample.

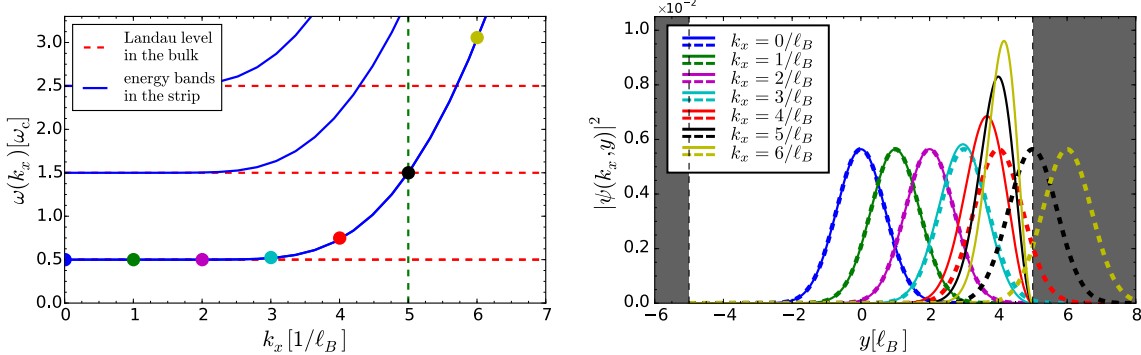

Figure 6: Left panel: Zoom of the dispersion bands in the IQHE. The vertical dashed green line is located at $k_x = L_y/2\ell_B^2$ marking the boundary of the sample. The colored dots indicate the eigen energies of the corresponding eigen wave functions. Right panel: Probability densities $|\psi(k_x, y)|^2$ of these eigen wave functions.

level. Deviations occur only in the tenth digit of the eigen energies. This is so because $k_x$ determines the position of the harmonic oscillator in $y$-direction, cf. Eq. (5). Closer to the boundaries, an upturn in energy occurs because the electrons feel the hard-wall in their vicinity. As pointed out above, the state $n = 0$ right at the boundary acquires the energy of the Landau level $n = 1$ because its wave functions corresponds to half a harmonic oscillator [40]. This relation is fulfilled up to the fifth digit thanks to the improved treatment of the second derivative at the boundary, see Fig. 3.

The gradual change of the eigen wave functions upon increasing $k_x$ is illustrated in Fig. 6. The colored dots in the left panel indicate the energies and the $k_x$ values of the eigen wave functions depicted in the right panel by solid lines of the same color. The dashed lines of the same color display the corresponding eigen functions in the bulk which remain of Gaussian shape throughout. Note the increase of the peak of the eigen functions in the strip geometry upon approaching the boundary (sequence red → black → yellow) because the electron cannot enter the hard-wall.

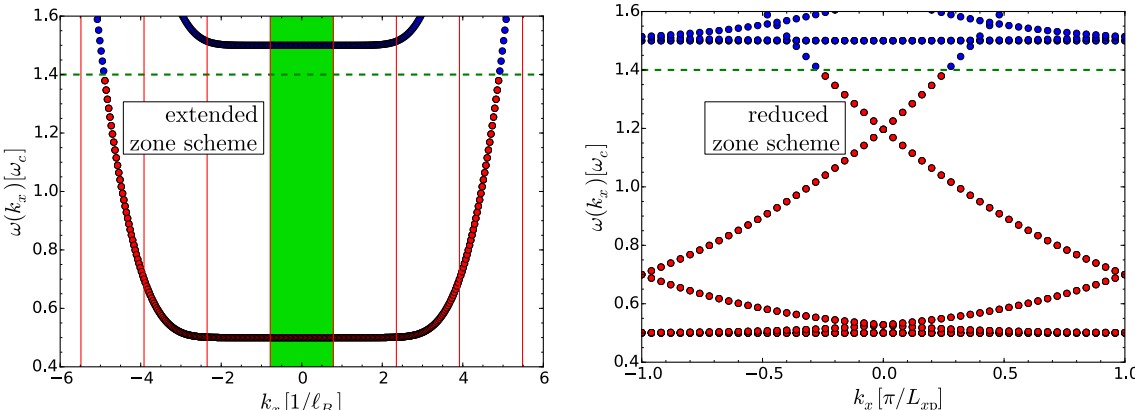

Figure 7: Left panel: Zoom of the lowest eigen energies in the extended zone scheme for $L_y = 10\,\mu\text{m}$ and $L_{x\text{p}} = 4.01\,\mu\text{m}$. (the deviation from $4\,\mu\text{m}$ is only due to the discretization). Red symbols corresponds to occupied states while blue symbols represent unoccupied states. The horizontal dashed green line indicates the chosen Fermi energy. The thin vertical red lines show boundaries of the corresponding reduced zone scheme. By backfolding the energies into the green shaded area one obtains the representation of the reduced zone scheme which is shown in the right panel.

## 3.2 Rectangular geometry

Next, we pass to the fully discretized model (14) for the sample without bays, see Fig. 4(b). This describes the same physics as the calculation in the previous subsection. Still, we present exemplary results in Fig. 7 for two reasons. The first one is to illustrate that this calculation indeed reproduces the results obtained previously on the mesh Fig. 4(a) with sufficient accuracy. Comparing the results from mesh (a) with those from mesh (b) in Fig. 4 we find that their eigen energies agree up to the fifth digit. Note that the calculation for mesh (a) requires to deal with a vector space of dimension of the order of 1000 while the calculation for mesh (b) requires to deal with a vector space with dimension of the order of $10^6$.

The second reason is to obtain results for the undecorated sample, i.e., without bays, as reference for the subsequent complete analysis. The main point is that the reduction of the translational invariance by considering the enlarged rectangular unit cell in real space of length $L_{x\text{p}}$ leads to a reduced zone scheme in $k_x$ space. The backfolded branches of the dispersion are shown in the right panel of Fig. 7. Since there is no real, physical reduction of the translational symmetry the backfolded branches display level crossings at the boundaries and elsewhere which are preserved as long as the physical translational symmetry is preserved. Hence the backfolded branches can be unfolded again to yield the extended zone scheme display in the left panel of Fig. 7. This shows the same results as were obtained directly by the previous calculation based on mesh Fig. 4(a), presented in Figs. 5 and 6.

For clarity, we have chosen in Fig. 7 to consider a quantum Hall sample of finite length. The length of the unit cell in real space is given by $L_{x\text{p}}$ and we fix the total number of these cells to $N_x = 50$. Of course, this value can easily be changed if needed. Hence, there are $N_x$ different momenta $k_x$ in the reduced zone scheme. They are multiples of $2\pi/N_x L_{x\text{p}}$ lying in the interval $\left[-\pi/L_{x\text{p}}, \pi/L_{x\text{p}}\right]$.

We want to focus on the filled lowest Landau level, i.e., filling factor $\nu = 1$. Due to the upturn of the lowest level upon approaching the boundaries of the sample this filling factor requires to occupy all states with energies just below the flat region of the second lowest level, see left panel of Fig. 7. However, in order to exclude any spurious effects of the energy levels of the second lowest Landau level we set the Fermi level to a value slightly below the flat band

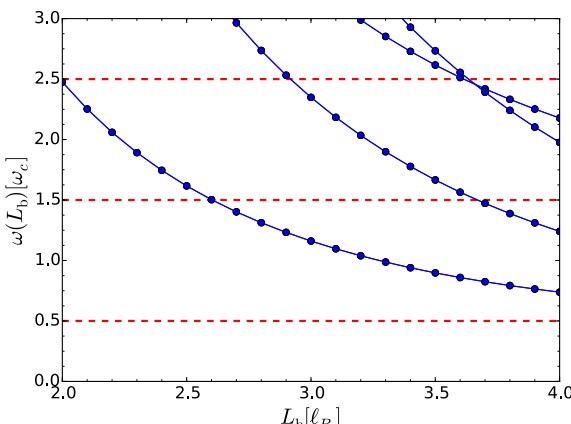

Figure 8: Discrete energy spectra of decoupled, i.e., isolated bays as function of their size $L_b$ are rendered by blue solid lines and blue symbols for $\ell_B = 1\,\mu\text{m}$. The horizontal red dashed lines indicate the equidistant Landau levels in the bulk for comparison.

of the Landau level $n = 1$, namely to $\epsilon_F = 1.4\omega_c$ as indicated by the green dashed line in Fig. 7. This allows us to distinguish unambiguously between occupied and unoccupied levels. This procedure helps to identify our quantity of interest, the Fermi velocity, i.e., the derivative of the dispersion with respect to $k_x$ at the Fermi level. The ensuing minor deviation of the filling factor $\nu$ from 1 is macroscopically irrelevant for large values of $L_y$.

### 3.3 Isolated bays

Before dealing with the complete system with bays coupled to the quantum Hall sample we determine the energy spectrum of isolated bays for later comparison. Note that we choose to consider quadratic bay for calculational simplicity. But the underlying physics does not require a particular shape of the bay so that samples decorated with circular bays will show the same physics at somehow modified quantitative parameters.

For considering the isolated bays we treat the mesh shown in Fig. 4 (c). The calculated energy spectrum as function of the length $L_b$ is plotted in Fig. 8. Having the classical cyclotron picture of circular electronic orbits in mind we choose $L_b = 2\ell_B$ as starting value. No smaller bay would allow for a classical circular orbit. As expected the energies are larger than the bulk Landau energies because the confinement due to the bays restricts the motion of the electrons. Accordingly, increasing $L_b$ lowers the energies because enlarging the bays reduces the influence of the confining potential.

The lowest eigen energy of the bay reaches the energy gap between the two lowest Landau levels at a bay size of $L_b \approx 2.6\ell_B$. Using the gate voltage $V_{g1}$ to shift the energies in the bays relative to the rest of the sample offers a possibility to tune a local mode in resonance to an edge mode. We will discuss this in more detail in the next subsection.

Adding the decoupled bay to the unit cell, i.e., considering the model shown in the panels (b) and (c) of Fig. 4 without any coupling yields the eigen energies provided in Sect. 3.2 plus the eigen energies of the bays which do not disperse at all (not shown). They appear as completely flat modes if plotted against $k_x$ due to their completely local nature in real space.

### 3.4 Quantum Hall sample with coupled bays

Now, we pass to the fully decorated sample where the bays are coupled to the 2D electron gas in the strip, i.e., we consider the mesh in Fig. 4(d). We switch on the coupling between the bays and the strip by gradually increasing the opening $L_o$ from zero to the maximum value $L_b$.

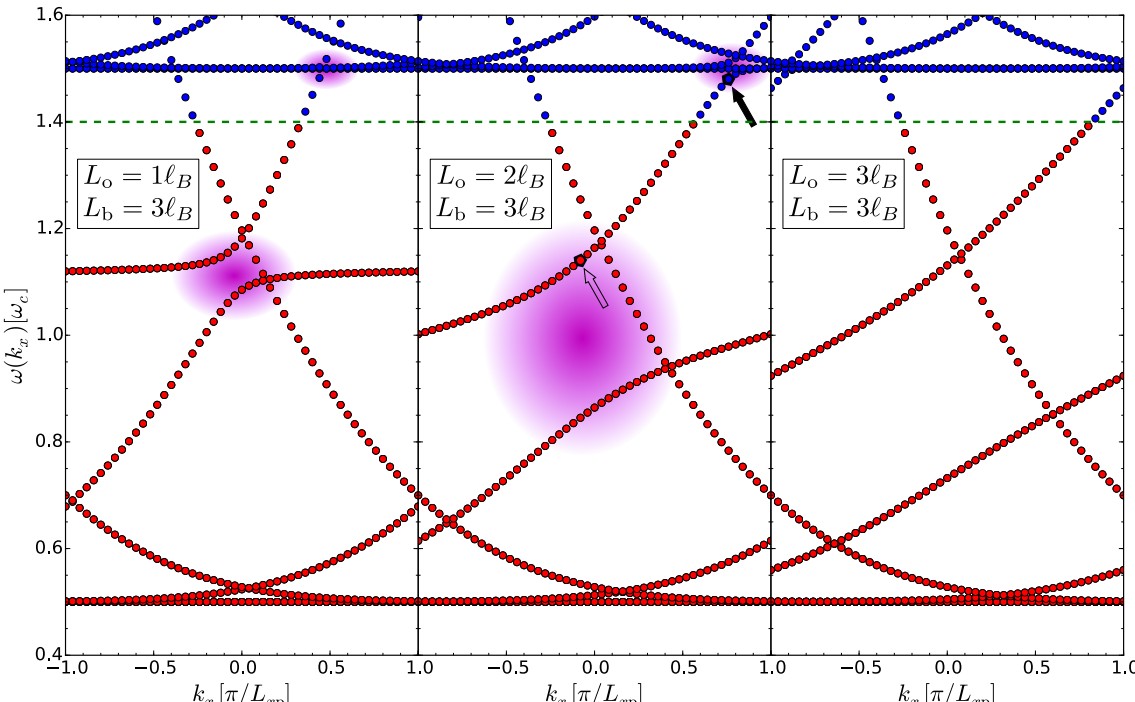

Figure 9: Energy spectra of the lowest eigen states of a quantum Hall sample with $L_y = 10\,\mu\mathrm{m}$, $L_{xp} = 4.01\,\mu\mathrm{m}$, and $\ell_B = 1\,\mu\mathrm{m}$. The left panel shows the case of weakly coupled bays because the opening $L_o$ is small. The middle panel shows a moderate coupling while the right panel a rather strongly coupled case because the opening $L_o$ is increased step by step. Red symbols correspond to occupied states while blue symbols depict unoccupied states; the dashed horizontal green line indicates the chosen Fermi level. The shaded areas highlight the locations of two avoid crossings due to the hybridization of local and dispersive modes.

The energy spectra are computed and tracked to understand how the coupling influences the eigen states in general and the edge modes in particular.

To this end, we depict three representative cases with openings $L_o = \{1\ell_B, 2\ell_B, 3\ell_B\}$ and a bay size $L_b = 3\ell_B$ in Fig. 9. They represent the cases of weak, moderate, and strong coupling of the bays. Upon coupling the bays to the quantum Hall sample, i.e., for $L_o \neq 0$, the eigen states of the bays and the strip start to merge. Energy crossings of local modes from the bays with dispersive edge modes in absence of any coupling turn into avoided crossings once the bays and the strip are coupled. This represents a clear finger print of level repulsion.

Inspecting the three panels, one realizes that only the right moving edge modes are influenced by the coupling of the bays. Only their energies depend on the degree of coupling, i.e., on the size $L_o$ of the opening. The left moving modes are spatially separated because they are localized at the other boundary of the sample without decoration. Hence they are influenced only exponentially weakly.

A nice example of the level repulsion between a (formerly) local bay mode and a dispersive, right moving edge mode is seen in the middle of the panels in Fig. 9 around $k_x = 0$. The relevant area is shaded in violet in the left and the middle panel. An example of a corresponding wave function is shown in the left panel of Fig. 10. In the right panel of Fig. 9 the avoided crossing is still present, but hardly discernible because the energies are already very different due to the strong coupling. In return, the left panel shows the character of an avoided level crossing most clearly because the coupling of the bays is still small and hence the hybridization between the bay modes and the strip modes is still small.

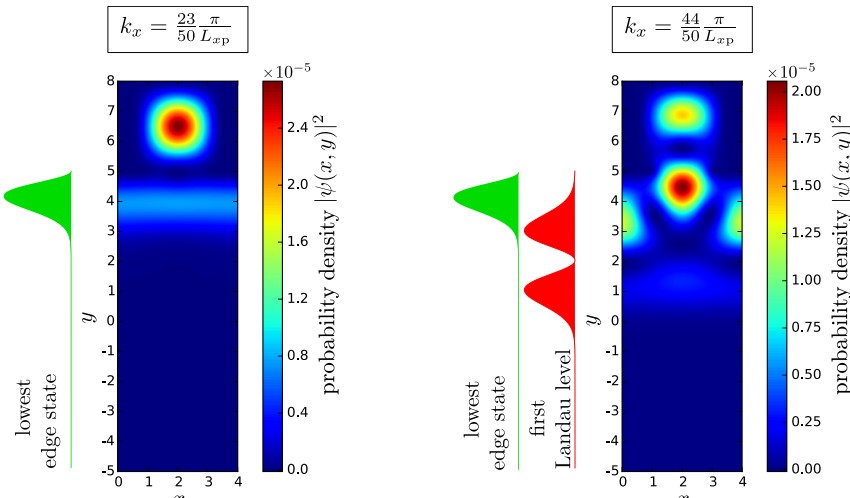

Figure 10: Probability density $|\psi_n(x,y)|^2$ of two eigen states influenced by the different avoid crossings. The left panel shows the hybridization between the edge state of the Landau level $n = 0$ with the local mode in the bay. The energy and momentum of this state are indicated in the middle panel of Fig. 9 by the open arrow. The right panel shows the weak hybridization of the edge state with the second Landau level $n = 1$ mediated by the local mode in the bay. The energy and momentum of this state are indicated in the middle panel of Fig. 9 by the filled arrow. The parameters of the geometry are $L_y = 10\,\mu$m, $L_{xp} = 4.01\,\mu$m, $L_b = 3\,\mu$m, $L_o = 2\,\mu$m, and $\ell_B = 1\,\mu$m.

Another, less obvious and thus surprising, origin of avoided level crossings between dispersive edge modes and local modes results from the breaking of the translational invariance and the concomitant backfolding. This mechanism induces hybridization between local Landau levels and edge modes. An example is indicated by a shaded area in the left panel at $k_x \approx 0.4\pi/L_{xp}$ and in the middle panel at $k_x \approx 0.8\pi/L_{xp}$ of Fig. 9. Clearly, the effect is weaker than the hybridization of edge modes and local bay modes. This is so because the coupling of edge modes and local Landau levels is a second order effect in the coupling of the bays to the strip. The bay modes are involved only indirectly by virtual processes, see also the right panel of Fig. 10 where an exemplary wave function is shown. Similar effects were also found in the IQHE where different edge modes start to mix with one other due to breaking the translational symmetry by a step potential [41].

To support the interpretations given above, we plot the probability density $|\psi(x,y)|^2$ for eigen states from the two avoided level crossings in Fig. 10. The left panel shows a state built from an edge mode and a local mode from the bays; its position in the energy spectrum is indicated by a solid arrow in the middle panel of Fig. 7. Clearly, the two constituents, the edge mode and the local mode in the bay can be seen.

The right panel Fig. 10 shows a state built from an edge mode, a local mode from the bays, and a the next higher Landau level $n = 1$; its position in the energy spectrum is indicated by a filled arrow in the middle panel of Fig. 7. Here, three states are involved and contribute to the eigen states as can be discerned nicely. The contribution of the local mode in the bay is much smaller than in the case shown in the left panel because it contributes only as virtual state mediating the breaking of the translational invariance.

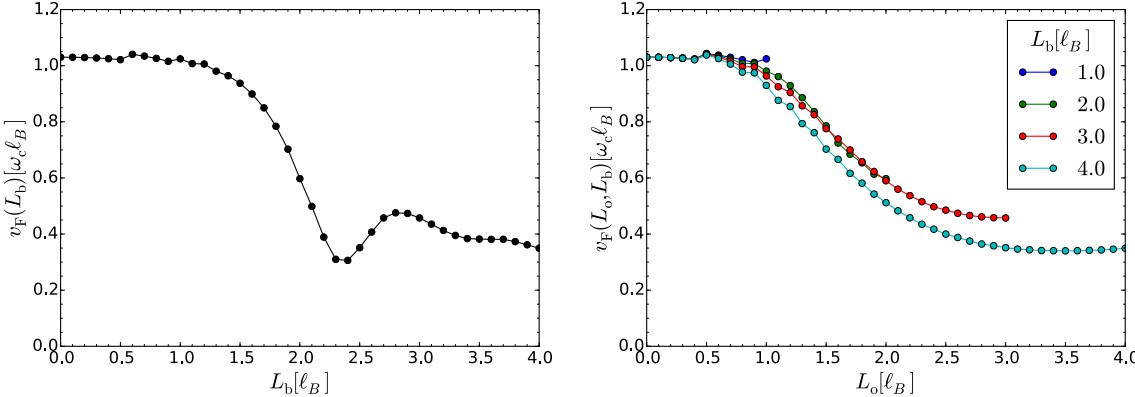

Figure 11: Left panel: Fermi velocity $v_F$ of the right moving modes as function of the bay size $L_b$ with at $L_y = 10\,\mu m$, $L_{xp} = 4.01\,\mu m$, $\ell_B = 1\,\mu m$, and $L_o = L_b$. Right panel: Fermi velocity $v_F$ as a function of the opening $L_o$ of the bays for various bay sizes $L_b$ with $L_y = 10\,\mu m$, $L_{xp} = 4.01\,\mu m$, and $\ell_B = 1\,\mu m$.

## 4 Tuning the Fermi velocity

In the previous sections we developed a detailed understanding of the energy spectra of quantum Hall sample decorated by bays. Our ultimate goal is to study whether and how the Fermi velocity $v_F$ can be tuned in such a decorated quantum Hall sample. We highlight that the Fermi velocity $v_F$ represents the group velocity of the coherent quantum mechanical propagation of electronic wave packets. It cannot be seen as classical propagation of electrons along the (longer) boundaries of the bays, see below. Here we present quantitative results for the Fermi velocity and its dependence on the parameters of the model.

First, we examine the dependence of $v_F$ on the size of the bays by increasing $L_b$ for maximally opened bays, i.e., for $L_o = L_b$. The results are shown in the left panel of Fig. 11. For maximally opened bays $L_o = L_b$ the dispersions display no flat region because the strong level repulsion induces sizable momentum dependencies of most modes, see right panel of Fig. 9. Thus no strong dependence of the Fermi velocity is expected in accordance with the left panel of Fig. 11. The complex interplay of many hybridizing levels makes it impossible to predict sizes for which parameter precisely $v_F$ takes its minimum value. However, the comparison of the left panel in Fig. 11 with Fig. 8 reveals that the Fermi velocity is indeed influenced when the local mode in the bay approaches the Fermi level, here $1.4\omega_c$, which is the case around $L_b = 2.6\ell_B$. Note that the Fermi velocity is generally reduced, roughly by a factor 2, once the local modes have come down in energy so that they reach the Fermi level.

The next parameter varied is the opening $L_o$ of the bay. The right panel of Fig. 11 shows the results for various bay sizes. Note that the opening cannot exceed the size of the bay, hence the curves stop at $L_o = L_b$. All curves follow the general trend that the Fermi velocity is lowered upon increasing the hybridization between local modes in the bays and the dispersive edge modes. This is achieved by increasing the opening $L_o$. An approximate reduction by a factor of 2 is achieved once the local energy levels from the bay come down in energy, i.e., for large enough $L_b$. This reduction is not very impressive; in addition, the geometry is fixed once the sample is grown and cannot be tuned on the fly.

The last dependence of $v_F$ on a geometric parameter, that we study, is the dependence on the distance between the bays, i.e., the size $L_{xp}$, of the decorated unit cell, see Fig. 2. One could imagine that a certain resonance phenomenon occurs for special values of $L_{xp}$. Generally, we expect that the influence of the decorating bays decreases upon increasing $L_{xp}$ because the fraction of decorated boundary decreases. Explicit results are shown in Fig. 12. Again, the

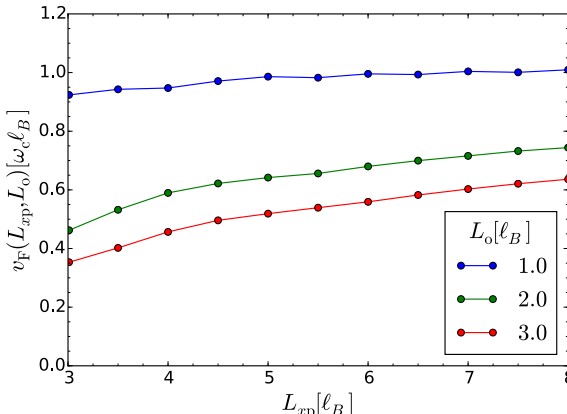

Figure 12: Fermi velocity $v_{\mathrm{F}}$ as function of the distance between the bays, i.e., $L_{x\mathrm{p}}$, see Fig. 2, for various bay openings $L_{\mathrm{o}}$ with $L_y = 10\,\mu\mathrm{m}$, $L_{\mathrm{b}} = 3\,\mu\mathrm{m}$, and $\ell_B = 1\,\mu\mathrm{m}$.

dependence of $v_{\mathrm{F}}$ is rather weak. The expected trend that larger $L_{x\mathrm{p}}$ reduces $v_{\mathrm{F}}$ less is clearly confirmed because the Fermi velocity approaches its undecorated value of about $1\omega_{\mathrm{c}}\ell_B$ upon increasing $L_{x\mathrm{p}}$. At small values of $L_{x\mathrm{p}}$ we retrieve a reduction of the order of a factor 2. But no resonance phenomena at particular values of the interbay distance are found. We attribute this to the fact that none of the local modes in the bay is truly in resonance with the edge modes

In order to identify a suitable tuning parameter we resort to the results gained for lattice models [28, 29]. Three ingredients are important for sizable changes of the Fermi velocity: (i) the local and the dispersive modes must be in (or close to) resonance. (ii) There must be a parameter to tune and to detune this resonance. (iii) The coupling of the modes should be rather small so that they are sensitive to being or not being in resonance.

Translating these conclusions back to the IQHE, it appears that we have to use the gate voltage $V_{\mathrm{g}1}$ to control the resonance between the local modes in the bays and the dispersive edge modes. It is obvious that one can shift the bay modes by changing $V_{\mathrm{g}1}$. An additional asset is that this can be done on the fly so that one disposes of a true control knob for the speed of signal transmission and hence for the delay time which can be turned while the signal processing is going on.

The opening of the bays should not be large because the coupling and hence the hybridization of the local and the dispersive modes should be rather weak. Thus we choose the rather small value $L_{\mathrm{o}} = \ell_B$ in Fig. 13. In this figure, we plot the dependence of the Fermi velocity on the gate voltage. For most values, the Fermi velocity does not deviate strongly from its value of about $1\omega_{\mathrm{c}}\ell_B$ in a sample without bays. But if the energy levels of the local modes in the bays approach the dispersive edge mode at the Fermi level they resonate and produce an avoided level crossing. In the region of the avoided level crossing the local mode and the dispersive one mix so that the formerly steep crossing of the dispersion through the Fermi level becomes flat. Hence the Fermi velocity is considerably suppressed. Note that the resulting resonance dips of $v_{\mathrm{F}}$ are rather narrow and can easily be used to (de)tune the velocity by moderate changes of the applied external gate voltage. In this fashion, changes of the Fermi velocity by factors 10 to 100 should be realizable, similar to what was found in lattice models [28, 29].

Comparing Figs. 13 and 14 one realizes the similarities of the curves. The width of the resonance dips is comparable if the openings $L_{\mathrm{o}}$ are the same, cf. Fig. 13 and the second lowest panel of Fig. 14.

Fig. 14 illustrates very clearly, that larger openings lead to significantly broader dips which are less deep. In return, smaller openings and thus less coupled bays lead to narrower dips

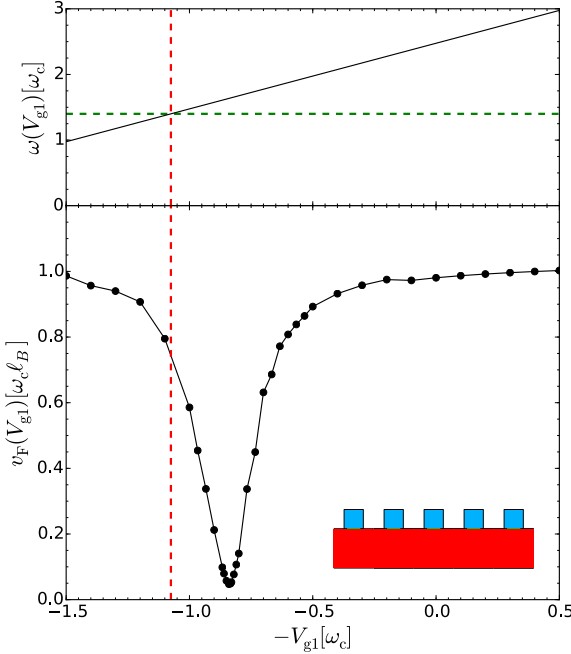

Figure 13: Upper panel: The horizontal dashed line shows the set Fermi level while the slanted solid lines depict the energy level in the isolated bays shifted by the gate voltage. The vertical red line is a guide to the eye to link the resonance visible in the upper panel to the strong response in the lower panel. Lower panel: Fermi velocity $v_F$ as function of the gate voltage $V_{g1}$ for $L_b = 2\,\mu m$, $L_y = 10\,\mu m$, $L_{xp} = 4.01\,\mu m$, $L_o = 1\,\mu m$, and $\ell_B = 1\,\mu m$.

with significantly lower residual Fermi velocity at the minimum. This minimum value of $v_F$ depends on how flat the dispersion of the hybridized modes remains as determined by the coupling strength: Weaker coupling implies better localized hybridized modes with flatter dispersion. Flatter modes allow for sharper dips to lower residual values of the Fermi velocity. Note that the reduction of the Fermi velocity can reach a factor of 100 for narrow bay openings. A classical interpretation of slower propagation due to the longer path along the boundaries of the bays would explain a factor 3.75 at best for $L_o = 0.5\,\mu m$.

The positions of the resonance dips depend on the energy levels of the modes in the bay so that the bay size influences them strongly. In the smaller bays studied in Fig. 13, the lowest bay level lies above the Fermi level so that the gate voltage has to bring it down in order to observe resonance. In the larger bays, studied in Fig. 14, the lowest bay level lies below the Fermi level while the second lowest above it. So Fig. 14 shows that several dips may occur, even for different signs of the gate voltages. All in all, it appears that the precise position of the dips is not at the resonance of the energy levels of the decoupled bays, but at slightly higher values of the gate voltage. We attribute this to the effects of the hybridizing couplings which shifts the local modes in the bays downwards in energy.

# 5   Conclusions

Topologically protected edge states possess many theoretically appealing properties. Still, avenues towards applications have not been followed by broad research. The recent proposal of tunable Fermi velocities in Chern insulators and spinful topological insulators for the realization of delay lines and interference devices is a step in this direction. The purpose of the present study was to show that no lattice models are required, but that semiconductor samples

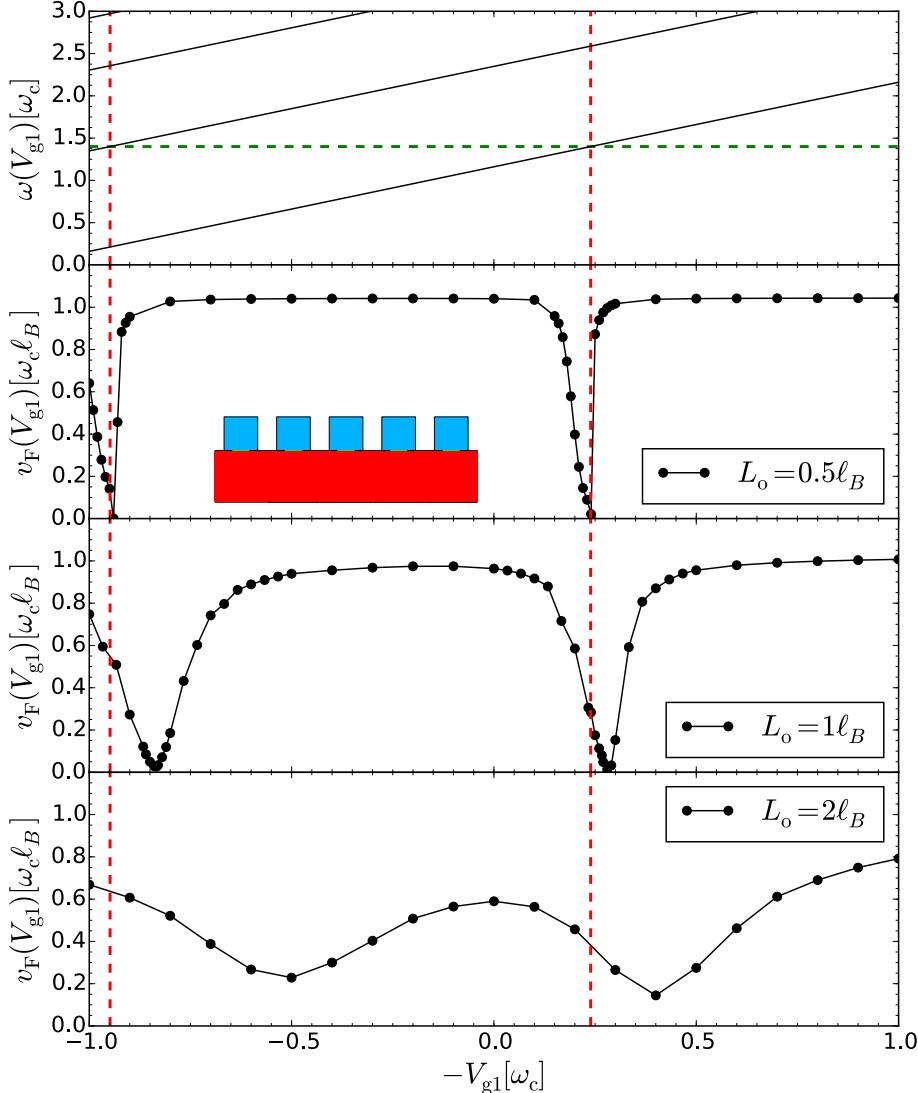

Figure 14: Upper panel: The horizontal dashed line shows the set Fermi level while the slanted solid lines depict the energy level in the isolated bays shifted by the gate voltage. The vertical red lines are guides to the eye to link the resonance visible in the upper panel to the strong response in the lower three panels. Lower three panels: Fermi velocity $v_F$ as function of the gate voltage $V_{g1}$ for $L_b = 3\,\mu m$, $L_y = 10\,\mu m$, $L_{xp} = 4.01\,\mu m$, $\ell_B = 1\,\mu m$, and the three different values of $L_o = 1\,\mu m$ as indicated.

with decorated boundaries show the same phenomena. This finding represents a substantial step forward towards realization because of the extremely high standard of designing and growing nanostructures for semiconductor devices.

We analyzed the dependence of the dispersion of the edge states in decorated quantum Hall samples on various parameters. The geometry of the sample sets the energy levels and partly the degree of coupling between the decorating bays and the bulk of the two-dimensional sample. Yet the geometric parameters do not allow for a fine-tuning of the Fermi velocity, let alone quick changes of it in the course of signal processing.

But gate voltages can achieve the wanted tunability. First, we found that the local levels in the bays should be close in energy to the Fermi level in the remainder of the quantum Hall sample so that the gate voltage applied to the bays does not need to shift them to a large extent. Second, the coupling between the bays and the rest of the sample should be rather

weak to have rather narrow and deep dips in the Fermi velocity if the local modes are tuned into resonance to the dispersive edge states. Then, the fundamental mechanism of mode mixing and level repulsion leads to weakly dispersive eigen modes crossing the Fermi level. This represents the key phenomenon for tunability.

Changes by up to two orders of magnitude appear possible. In our calculations, the degree of coupling is a geometric parameter. In practice, we propose to make it tunable as well by additional gate electrodes which modify the width of the opening of the bays, cf. Ref. [35].

The calculations are based on discretizing the sample in real space and mapping it to a tight-binding type of model. For fine enough meshes, reliable results valid for the continuum case are obtained as we could verify by comparison to analytic bulk solutions. We increased the complexity of the considered geometry step by step in order to gain a reliable understanding of the occurring physical phenomena. The approach is flexible enough to be adapted to various geometries. We considered quadratic bays, but any other shape is possible as well, but only small, quantitative changes are expected. Here the focus was on a proof-of-principle calculation to show that the anticipated physics takes indeed place in the integer quantum Hall effect.

In view of experimental realizability, some aspects must be kept in mind. First, the neglected interaction between the electrons may lead to the formation of certain charge modulations at the boundary. On the one hand, it is established that compressible and incompressible stripes form close to the boundaries [42]. The incompressible stripes may hinder the propagation of signals. On the other hand, if the filling is tuned just below filling factor $\nu = 1$, we expect that this effect is avoided because no incompressible stripes are formed at the edges. The final clarification, however, can only be reached by an experimental study.

For concreteness, we showed calculations for $\ell_B = 1\,\mu m$. This value corresponds via $B = \hbar/(e\ell_B^2)$ to a magnetic field of 0.66mT and to a electron density of $3.2 \cdot 10^7 \mathrm{cm}^{-2}$. Both values are very small compared to the values in generic quantum Hall setups which have magnetic fields and electron densities higher by about a factor $10^4$. Thus, for realization one has to look for systems with high mobility at much smaller electron densities or to make the geometric structures of the sample smaller, e.g., a factor 5 in linear dimensions yields a factor 25 in the electron density and in the magnetic field.

An interesting alternative to standard semiconductors is the quantum Hall effect in graphene. The relation between magnetic length $\ell_B$ and magnetic field is the same [43–47], but the relevant electron density $n$ is measured relative to the semimetal so that small values are easily realized. Due to the density-of-states linear in energy one has $n \propto \epsilon_F^2$. Furthermore, due to the perfect lattice structure a high mobility can be expected. So the promising aim is to create non-trivial boundaries with bays on the scale of 10 to 1000nm in graphene.

In conclusion, an experimentally realizable topological phase, the integer quantum Hall effect, allows for tunable Fermi velocities if its edges are appropriately decorated. Gate voltages can serve as control parameters for tuning. These findings should encourage further research to realize such systems on the laboratory level to ultimately pave the way towards real devices.

As an outlook we want to emphasize that the presented finding can be extended in several ways as has been done for lattice models [29]. The detrimental effects of disorder can be included to study the robustness of the observed effects. Such investigations will help to understand with which accuracy an experimental realization has to be grown in order to be able to observe the predicted effects. Without doubt, this constitutes an essential step toward applications.

Second, our findings can be extended to spinful models without conceptual difficulties. If the spin is subject to spin-orbit coupling the chiral edge modes will generically become helical modes which opens up the promising field for applications in spintronics, for instance realizing switchable spin diodes. Thus, many tantalizing research projects lie ahead.

## Acknowledgements

We acknowledge useful discussions with Manfred Bayer, Axel Lorke, Bruce Normand, and Dirk Reuter.

**Funding information**   One of the authors (MM) gratefully acknowledge financial support by the Studienstiftung des deutschen Volkes. This work was also supported by the Deutsche Forschungsgemeinschaft and the Russian Foundation of Basic Research in the International Collaborative Research Center TRR 160.

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
