# Peer review of "Tunable dispersion of the edge states in the integer quantum Hall effect"

_SciPost Physics, doi:SciPost Phys. 3, 032 (2017)_

## Round 1 · Referee Report · Anonymous · 2017-9-20

Strengths
The paper is very well organized and contains specific details of the calculations.
Weaknesses
See "requested changes"
Report
This manuscript provides details of tunable delay lines which work in the integer quantum Hall regime. I find this effort highly original and important as there are relatively few experimental realizations of devices which rely on the so called topological property or the presence of the edge states.
Requested changes
I found only one issue which needs clarification. Past page 15, the authors discuss edge states coupled to a series of bays. The authors have explained clearly that the Fermi velocity will be slowed down substantially in the region where the edge couples to one bay. However, in between such coupled regions there are other regions, each of length L_xp-L_o which I believe are not coupled to the bays, therefore the Fermi velocity is unchanged here. I therefore suspect that in Fig.11-14 the average velocity is plotted. As such, such an average velocity will indeed describe the proposed delay line.
However, from the text of the manuscript, is was not clear if the authors did indeed compute such an average speed. Should the average speed decrease all the way to zero at certain gate voltages in Fig.14?
So I am asking the authors to clarify if in Fig.11-14 they intended to discuss the average velocity.
Efrat Shimshoni on 2017-10-16 [id 182]
The manuscript presents an idea that looks nice in principle: to use a Quantum Hall system in the IQHE regime with edges decorated by weakly couples bays as a basis for designing devices such as a delay line, or an interferometer with tunable dynamical phases on each arm. The principal idea is to tune the level of hybridization with local states in the bays (e.g. by gate voltage), thus allowing to control the Fermi velocity of chiral edge states. The authors perform a numerical study on a lattice system with specific geometry, and demonstrate a proof of principle that substantial tenability of edge states velocity can be achieved.
This is certainly a nice original idea, which suggests a possible practical application of quantum Hall system beyond the setting of resistance standard. My main concern about the study is that the effect of Coulomb interactions is entirely ignored. The authors restricted themselves to the IQHE to justify their use of a non-interacting model. However, it is well-known that even in the integer regime, the role of long-range Coulomb interaction can lead to effects such as edge reconstruction - especially in a complicated geometry as proposed by the authors, and when smooth edges are involved (as typically is the case in semiconductor devices where such patterns can be realized).
I therefore would recommend that the authors address this point before approving publication in SciPost. It is quite possible that the predicted behavior discussed in the manuscript is nevertheless qualitative correct; ultimately, an experimental realization of the proposed setup would be required for a convincing proof-of-principle.
Author: Götz Uhrig on 2017-10-16 [id 183]
(in reply to Efrat Shimshoni on 2017-10-16 [id 182])Dear Efrat Shimshoni,
many thanks for the positive judgement that we present "a nice original idea, which suggests a possible practical application of quantum Hall system beyond the setting of resistance standard." Concerning the effect of long-range Coulomb interaction, we agree that this issues deserves to be discussed, e.g., based on the paper by Chklovskii et al. PRB 46, 4026 (1992). Our present view is that the problems of (compressible and incompressible) stripes at the edges can be avoided by focussing at \nu=1 and slightly below this value. Then one should have a compressible region at the boundaries which essentially behaves as proposed, perhaps with some renormalized boundary potential.
In any case, we agree that clarity on this issue can only be reached by experimental verification.
Best regards

---

## Round 1 · Referee Report · Anonymous · 2017-10-12

Strengths
This paper paper reports on a numerical study of edge states in a quantum Hall system coupled to a periodic array of "bays" with tunable bound energy states. By adjusting these energetically the authors convincingly show that the Fermi velocity of the system can be very substantially decreased from its unperturbed value. The physics is very understandable and appealing, and the numerical results quite convincing.
Weaknesses
I do not notice any significant weaknesses.
Report
I found the paper to be well-written and potentially quite useful, particularly to people wishing to use these methods for studies of quantum Hall systems in structured environments. The agreement between the numerical results and what one expects upon some thought is quite satisfying.
Requested changes
I have no requested changes.

---

## Editorial Decision

published